# Science in motion: A qualitative analysis of journalists' use and perception of preprints

**Alice Fleerackers**[1]*, **Laura L. Moorhead**[2]*, **Lauren A. Maggio**[3], **Kaylee Fagan**[2], **Juan Pablo Alperin**[4]

**1** Interdisciplinary Studies, Simon Fraser University, Vancouver, British Columbia, Canada, **2** Journalism, College of Liberal and Creative Arts, San Francisco State University, San Francisco, California, United States of America, **3** Department of Medicine, Uniformed Services University, Bethesda, Maryland, United States of America, **4** Publishing Program, Simon Fraser University, Vancouver, British Columbia, Canada

* afleerac@sfu.ca (AF); lauralm@sfsu.edu (LLM)

## Abstract

This qualitative study explores how and why journalists use preprints—unreviewed research papers—in their reporting. Through thematic analysis of interviews conducted with 19 health and science journalists in the second year of the COVID-19 pandemic, it applies a theoretical framework that conceptualizes COVID-19 preprint research as a form of *post-normal science*, characterized by high scientific uncertainty and societal relevance, urgent need for political decision-making, and value-related policy considerations. Findings suggest that journalists approach the decision to cover preprints as a careful calculation, in which the potential public benefits and the ease of access preprints provided were weighed against risks of spreading misinformation. Journalists described viewing unreviewed studies with extra skepticism and relied on diverse strategies to find, vet, and report on them. Some of these strategies represent standard science journalism, while others, such as labeling unreviewed studies as preprints, mark a departure from the norm. However, journalists also reported barriers to covering preprints, as many felt they lacked the expertise or the time required to fully understand or vet the research. The findings suggest that coverage of preprints is likely to continue post-pandemic, with important implications for scientists, journalists, and the publics who read their work.

## Introduction

COVID-19 has changed many aspects of how health research is communicated. Among these changes has been a surge, both within and beyond the scholarly community, in the use of *preprints*, research papers posted online before formal peer review [1]. Preprints are useful for rapid information sharing in outbreak contexts [2], as they allow researchers, by circumventing the often lengthy peer-review process, to share findings and build on one another's work more quickly than would otherwise be possible [3]. Yet, their use in the health and biomedical sciences has historically lagged behind uptake in other fields [4]. This hesitance may be due to the potential danger that unverified findings could receive premature media coverage [5] and,

**Data Availability Statement:** News story data can be accessed via the Harvard Dataverse at https://dataverse.harvard.edu/ with the doi: 10.7910/DVN/KK6T86. Interview guides can be accessed via the

Open Science Framework at doi: 10.17605/OSF.IO/V98NQ. Interview transcripts cannot be shared publicly because they contain information that could be used to identify participants.

**Funding:** This research is supported by a Social Sciences and Humanities Research Council of Canada (SSHRC) Insight Grant (#453-2020-0401; JPA) and a SSHRC Joseph Bombardier Doctoral Fellowship (#767-2019-0369; AF). There was no additional external funding received for this study. The funders had no role in study design, data collection and analysis, decision to publish, or preparation of the manuscript.

**Competing interests:** The authors have declared that no competing interests exist.

ultimately, mislead audiences. This fear may also explain why journalistic preprint coverage has often been discouraged within and outside of journalism [6, 7].

The urgency of addressing the pandemic, however, seems to have outweighed this risk for many journalists and researchers, including those covering or in health and biomedical fields. As early as January 2020, scientists across the disciplinary spectrum began posting preprints in numbers not seen before [8–10], and COVID-19-related preprints soon surpassed those on other subjects in terms of uploads, views, downloads, comments, and citations [11]. Yet, it was not only the scholarly community that increased its use of preprints to meet pandemic demands. In the absence of relevant peer-reviewed research, media coverage of COVID-19 preprints saw a parallel surge [11], with some journalists reporting on them for the first time [12]. This surge was not restricted to the domain of specialized health and science reporting but instead engaged a wide range of media outlets, including major generalist outlets, such as *The New York Times* and *The Guardian* [13, 14]. On the one hand, this broad uptake may have benefited audiences, as many of the most highly covered preprints provided insights into key public health issues such as disease transmission, intervention, and treatment [11]. However, some of the longstanding fears associated with premature media coverage have also played out, with several flawed or biased preprints gaining considerable media attention [15, 16].

In light of this problematic media coverage, scholars have called for more research examining the reasons behind journalists' deviation from professional norms when covering preprints [17] and, more broadly, for "Further consideration . . .about the place of preprints and how to counter their possible harm on public discourse" [18]. This study addresses these calls through an exploration of the use of preprints in health and science news from the perspective of journalists. In particular, it seeks to understand journalists' motivations to cover preprints and the degree to which they have adopted novel reporting practices to mitigate the potential risks of doing so—a topic which scholars have only begun to address [19]. We conducted semi-structured interviews, which we analyzed using a theoretical framework of *post-normal science communication* [20], to examine whether the way in which journalists find, verify, and communicate preprint research represents a departure from "normal" science journalism and, if so, whether this departure is likely to persist post-pandemic.

## COVID-19 as a post-normal science (communication) context

We conceptualize journalists' preprint use during the pandemic as a response to *post-normal science* (PNS) [21, 22], defined by four features, all of which apply to the COVID-19 context: i) high levels of scientific uncertainty, ii) science policy considerations that involve values (not just evidence), iii) high relevance to society, and iv) an urgent need for political decision-making. Such contexts challenge the norms of science, pushing researchers to consider and engage with an "extended peer community" that includes policy makers, journalists, and members of the public. Journalists and other science communicators must also adapt their norms and practices in PNS contexts; as Brüggemann and colleagues write, "These post-normal situations, combined with the changing media environment and a polarized society, shape and challenge the professional roles and norms that underlie their communication practices." [20]

Scholars have documented several practices that science journalists "normally" use to find, verify, and communicate research, summarized in Table 1. These studies suggest that journalists apply a mix of passive and active strategies to find research studies; rely on triangulation, quality assessments, and the opinions of outside experts to verify them; use quotes from scientists to add context to their coverage; and strive to communicate research simply and objectively.

Table 1. Normal and post-normal science journalism practices described in previous research.

| Normal science | | Post-normal science |
|---|---|---|
| **Find research** | *Passive methods*: receiving press releases, pitches, PR materials, news alerts [23–25]<br>*Active methods*: accessing research through academic search tools, social media, specific journals, or contact with researchers [23, 26, 27] | *Accurately portray "tensions and dissensions"* within science by incorporating perspectives from researchers and stakeholders on all sides of the issue; describing expert concerns; communicating scientific uncertainties [28, 29] |
| **Verify research** | Assess *quality-related factors*, such as "whether the research was sound, whether the source was reputable" [26]<br>Corroborate or critique study claims through *commentary from unaffiliated experts* [23, 25]<br>*Triangulate* statistics and findings by comparing them with those of other credible sources [25] | *Engage audiences* by encouraging public comments/feedback, making data and information directly available, acting as a "dialogue" or "knowledge broker" [28–30]<br>*Interpret science* by putting research into context, describing the process of science (not just results); widening perspectives on polarized debates; highlighting policy implications; considering long-term outcomes [20, 29–31] |
| **Communicate research** | *Use expert quotes* from study authors and unaffiliated researchers to set context, legitimize research, establish a sense of balance, articulate societal implications [32]<br>*Translate or simplify science* to make it more understandable to lay audiences [20, 33]<br>*Strive for objectivity* [20, 34] in all reporting | *Strive for reflexivity;* call objectivity into question; incorporate subjective feelings/views; be transparent about values; frame differing perspectives as context-specific rather than competing [20, 29, 31]<br>*Advocate* for common goods and/or social transformations [20, 31]<br>Critically discuss *solutions*, rather than simply reporting what is wrong [31] |

Scholarship on PNS journalism is more limited and has focused primarily on *communication practices* rather than on how journalists find or verify research. This body of literature (summarized in Table 1) proposes that journalists should strive to bring reflexivity into their reporting, communicate tensions or uncertainties, and highlight (or even advocate for) potential solutions rather than simply reporting on problems. It also suggests that journalists should contextualize new research, describe the process of science, and actively engage audiences in dialogue. However, scholarship on PNS journalism described above is largely prescriptive rather than descriptive, outlining how journalists *should* communicate about PNS and not how they actually do so. It has also focused primarily on climate science, a limitation given that the norms that emerge from PNS situations may differ across contexts [20]. This research contributes to filling both of these gaps by extending the PNS framework to a novel context (COVID-19 preprints) and by documenting journalists' practices for covering this post-normal research.

## Research questions and objectives

The practices, roles, and norms that emerge during PNS situations can either complement or replace existing ones [20]. This can be seen in the solutions scholars and journalists have proposed to mitigate the potential risks associated with preprint media coverage, which include consulting unaffiliated experts [35], assessing study quality with a critical eye [36], and "emphasizing the preliminary nature of conclusions" [18]. Rigorous fact checking, working closely with study authors, and using independent sources to validate research findings have also been identified as important protective measures [37–39], as has building awareness among journalists and their audiences about the nature of preprints [3]. Although many of these recommended practices, such as fact checking or consulting unaffiliated experts, are simply "basic science journalism principles" [40], others, such as labeling papers as unreviewed or helping audiences understand the process of scholarly publishing, mark a departure from traditional journalistic practice. That is, the proposed solutions for reporting on preprints represent a combination of "normal" and "post normal" activities.

Perhaps-because some of these activities are post normal, their uptake among journalists has been uneven. Studies find that media stories mentioning COVID-19-related preprints early in the pandemic inconsistently described these studies as preliminary, unreviewed, in

need of verification, or a "preprint" [13, 17, 41]. Those stories that *do* make the preprint status of the research clear tend to offer only a brief explanation (or none at all) of what the term *preprint* means or how it relates to the larger academic publishing system [14, 17]. However, while these results shed some light on what audiences may encounter in preprint news coverage, they fail to capture what might be going on behind the scenes. That is, it remains unknown whether and how journalists apply other, less visible recommended practices for covering preprints, such as critical evaluation, consultation with outside experts, or use of outside sources to verify results. It is also unclear whether journalists' coverage of COVID-19 preprints during the pandemic is an artifact of the crisis or evidence of a larger shift in journalism practice. This research aims to help fill these gaps by addressing the following research questions:

RQ1. What benefits and risks do journalists consider in deciding whether to cover preprints?

RQ2. What practices do journalists use to find, verify, and communicate the preprints they cover?

RQ3. How has the COVID-19 pandemic affected journalists' use of preprints?

## Materials and methods

We conducted a qualitative interview study informed by a constructivist paradigm using qualitative description [42]. Qualitative description was selected for its utility as an appropriate methodology when interviewing those who directly experienced the phenomenon of study and when the researchers seek to understand "why, how and what questions about human behavior, motives, views, and barriers" [43]. This study is part of a larger research project examining the journalist-scientist relationship; only sections of interviews directly related to preprints or peer review were analyzed. The Simon Fraser University Research Ethics Board (# 30000244) and the San Francisco State Institutional Review Board (#2021175) exempted the project from further review. All participants provided written consent to participate in this research. The authors engaged in this research have backgrounds in education, journalism, medicine, and scholarly communication. NA, KF, AF, and LLM have worked in journalism. All of the authors have posted preprints.

### Sample

All interview participants (described below) worked for one of the following outlets: *The Guardian* (science section), HealthDay, IFL Science, MedPage Today, *News Medical*, *New York Times* (science section), *Popular Science*, and *Wired*. These publications were selected for their focus on science and health news, as well as their reach and popularity with readers in Canada, Europe, and the US. These outlets also represent the changing media landscape [44, 45], as they include the science sections of traditional, legacy news organizations (i.e., *The Guardian*, *New York Times*) and historically print-only science magazines (*Popular Science*, *Wired*) as well as digital native health sites (News Medical, MedPage Today) and science and health blogs (HealthDay, IFLScience).

### Participants

The 19 health and science journalists who participated in this study reported on research for one or more of the previously mentioned eight outlets. We identified journalists from these publications first by collecting all the stories available through the outlet's RSS feed or, if a feed was unavailable, through the Twitter timeline of the official account that posted a link to every story. Using these two methods, we identified stories published in the corresponding sections

between March 1 and April 30, 2021. We then read each story for mentions or links to research (both preprints and peer reviewed) and saved the accompanying bylines. Scripts used to identify and save stories are openly available [46].

## Recruitment

LLM randomly ordered the sampled stories in Google Sheets and recruited from bylined authors, top to bottom, from the ordered list; bylines appearing to be from organizations (e.g., American Heart Association News) and politicians were excluded. LLM gathered contact information from publicly available sources (e.g., outlet masthead or contact listing, personal website). KF emailed potential participants up to three times. 19 journalists from seven of the eight publications agreed to be interviewed (see Table 2 for participant characteristics). Recruitment and interviews occurred between July and November 2021.

## Interviews

KF conducted semi-structured interviews of journalists via Zoom. Participants were asked about their professional experience with reporting on preprint and peer-reviewed research and how the pandemic had affected that experience and their views on the use of preprints. The interview guide is available online [47]. Interviews lasted between 10 and 47 minutes, with most averaging about 35 minutes. All interviews were recorded and then transcribed by a third-party company; transcripts were de-identified prior to analysis.

**Table 2. Characteristics of journalists who participated in interviews (n = 19) about use of preprints.**

| Journalist | Primary outlet | Primary outlet description | Employment status (staff or freelance) | Years in journalism |
|---|---|---|---|---|
| J1 | IFLScience ("I fucking love science" IFLS) | UK-based science blog | On staff | 6 |
| J2 | Popular Science | US-based science news and feature publication | Freelance | 7 |
| J3 | Popular Science | US-based science news and feature publication | Freelance | 2 |
| J4 | Popular Science | US-based science news and feature publication | Freelance | 6 |
| J5 | Wired | US-based science, technology and culture publication | Freelance | 33 |
| J6 | Medpage Today | US-based medical news service provider | On staff | 1 |
| J7 | Medpage Today | US-based medical news service provider | Freelance | 25 |
| J8 | Popular Science | US-based science news and feature publication | Freelance | 1–2 |
| J9 | Wired | US-based science, technology and culture publication | On staff | 4 |
| J10 | IFLScience | UK-based science blog | On staff | 8 |
| J11 | Popular Science | US-based science news and feature publication | On staff | 25 |
| J12 | Medpage Today | US-based medical news service provider | On staff | 6 |
| J13 | The Guardian | UK-based news and media publication | Freelance | 10 |
| J14 | The Guardian | UK-based news and media publication | On staff | 14 |
| J15 | Wired / Ars Technica | US-based science, technology and culture publication / US-based technology, science and political news publication | Freelance | 8 |
| J16 | Wired | US-based science, technology and culture publication | On staff | 28 |
| J17 | Popular Science | US-based science news and feature publication | On staff | 7 |
| J18 | New York Times | US-based daily news publication | Freelance | 9 |
| J19 | The Guardian | UK-based news and media publication | On staff | 3 |

NB. To protect journalist identities, education information is reported in aggregate only. All 19 participants had received at least one educational certificate or degree, with all but 2 reporting that they had a bachelor's degree or higher in a social sciences and humanities (SSH) field (n = 17). Many journalists also had training in a Science, Technology, Engineering, or Math (STEM) field (n = 6), with 3 journalists stating that they had attained a graduate degree in this area. Finally, 8 participants reported having received professional journalism education through a certificate, bachelor's, or master's program.

## Data analysis

Our analysis was guided by Brüggemann et al.'s framework for analyzing and understanding post-normal science communication [20]. This framework comprises five analytical steps, which we address as follows:

1. Classify whether the situation has post-normal characteristics (Literature Review);

2. Document how actors (e.g. journalists) are reacting to the situation (Method, Results);

3. Compare these reactions to what would be expected in a "normal" context (Literature Review, Discussion);

4. Explain what might be causing the divergences (Discussion); and

5. Consider the societal implications of these emerging norms (Discussion).

We selected thematic analysis as our method for *documenting how actors are reacting to the situation* due to its overall flexibility and for its utility in identifying experiences, perspectives, and behaviors across a data set [48, 49]. Interview transcripts were de-identified, then inductively analyzed using Braun and Clarke's steps of thematic analysis [50, 51], which allowed us to identify, examine, and report patterns in how journalists viewed and used preprints in their work. This process began with three researchers (AF, LAM, LLM) independently undertaking a close line-by-line reading of the first 12 transcripts to familiarize themselves with the data. Next, the authors independently identified initial codes, example quotes, and working definitions of the codes relevant to the research questions and informed by the literature on "normal" science communication, discussed above. All code data were managed in Google Sheets and shared amongst the research team during several collaborative video conference discussions. Guided by these discussions, and informed by the research questions and theoretical framework, a single researcher (AF) reviewed each author's codes, identified patterns and areas of overlap, and synthesized the most relevant and common themes into a series of tables comprising working theme labels and exemplar quotes. At this point, the team reviewed these thematic tables, added comments and suggestions, then met again to discuss the findings and whether *sufficiency* (i.e., the point at which the collected data from participants enables researchers to answer the research question) [52, 53] had been met.

The team agreed that data collection should continue, so an additional 7 interviews were conducted, transcribed, de-identified, and coded. Based on the collective transcripts, the team agreed that data and analytical sufficiency had been achieved.

## Results

Based on interviews with 19 health and science journalists, ranging in duration from 10 to 47 minutes and representing seven news publications (see Table 2), we identified a variety of themes to answer our research questions. Below we report these themes in relation to the research questions they address. For a summary of themes and representative quotes see Table 3.

### RQ1. What benefits and risks do journalists consider when deciding to cover preprints?

Across the interviews, most journalists described the decision to cover preprints as a careful *risk-benefit analysis*. This was true both in general (i.e., when deciding whether to cover preprints at all), as well as on a case-by-case basis (i.e., when deciding whether to cover or cite a specific preprint). At the heart of this decision was a consideration of *audience needs*, where

**Table 3. Themes identified in interviews with journalists (n = 19) about their use of preprints.**

| Theme | Definition | Example quotes |
|---|---|---|
| **RQ1: What benefits and risks do journalists consider in deciding whether to cover preprints?** | | |
| **A risk-benefit equation** | Journalists weighed the risks of covering preprints against potential benefits for the public; audience needs were central this calculation | The calculation is: 'Do we think that the audience needs to hear the story now or can they wait six to eight weeks. . . for the story to be peer reviewed?' And most of the time we think the wait is important and we tend to for the vast majority to avoid picking preprint [J1] |
| **Accessibility (Benefit)** | Preprints were valued because they were free to access. This benefit was both practical (i.e. easier for journalists) and ethical (i.e. a belief that knowledge should be free) | Preprints, it's easy, because they're freely available. . . Luckily, for Covid, a lot of things are open access. Maybe this is the future of science. It should be. But for now, we have to manage as we can. [J12] |
| **Timeliness (Benefit)** | Preprints were valued because they allowed more timely access to relevant research than was possible through peer reviewed research | [As journalists,] our allegiance is to our readers, and getting accurate but timely information to readers. . . When people are dying, you gotta get things going a little bit. And so that's, I think, what we've seen in the last year, in this argument over preprints. [J5] |
| **Potential to misinform (Risk)** | Covering preprints was seen as risky because unreviewed results could turn out to be false or flawed, contributing to misinformation | There was a study that was bad about running, how doing more running caused you to expel the virus. That was totally bunk. It was one of the things that like after a week of it making the rounds everywhere, everyone realized, 'Oh, wait. That wasn't true at all.' [J2] |
| **Difficult to verify (Risk)** | Lacking the expertise, resources, or skills needed to verify preprints was described as a major challenge | Just because I have a Ph.D. in [the field] doesn't mean that I have in most cases the right expertise to look at the paper and perfectly judge it. This is why we tend to favor covering that at peer reviewed. [J1] |
| **RQ2: What practices do journalists use to find, verify, and communicate the preprints they cover?** | | |
| **Active strategies (Finding preprints)** | Journalists actively searched for new preprints, most often direct from the servers themselves | With preprints, you tend to have to just go on, like, the preprint websites and kind of just sift through it and, like, see. [J11] |
| **Passive strategies (Finding preprints)** | Journalists received preprints from other sources, such as PR services and from authors themselves | [Preprint research] only comes into my life usually when I'm already sort of interviewing someone and they say, well, we have this piece that's out for peer review. [J11] |
| **Extra skepticism (Verifying preprints)** | Journalists felt that an added layer of skepticism was needed when verifying preprints; often linked to a trust in peer review as a quality control mechanism | There's another level of skepticism that should go into reporting on preprints, because there's one less safety net, basically, that the research has gone through [J8] |
| **Critical reading (Verifying preprints)** | Journalists verified preprints by reading studies with a critical eye, asking critical questions of the methods, sample, analyses, and findings; this practice was seen as a standard aspect of any science journalism | We're using the same toolkit, our toolkit for looking at a paper and evaluating its newsworthiness. We're like, 'Okay. Well, is this a good paper? Is the science good? Do the statistics make sense to us? Do the results actually answer the question that it says it answers, and what's left out?" We ask those things of formally-published—you know, if it comes out in *Science*. We ask those questions, too, because sometimes the answer is 'No' and sometimes the answer is like, 'Actually, this does seem dicey' [J16] |
| **Triangulation (Verifying preprints)** | Journalists verified preprints by comparing findings to information from other trusted sources, such as peer reviewed papers, experts, or other preprints | If we can find some article, in that case, we look more at similar work in the literature to back up some of the claims [J1] |
| **Do your own peer review (Verifying preprints)** | Journalists used outside scientists to verify preprints—to comment on methods, results, and significance in a process resembling scholarly peer review | In my opinion you can't do an unsourced preprint coverage. Like you need to ask like 10 doctors or 10 you know epidemiologists or 10 whatever relevant you know specialists there are like, "What did you think of this" and then you need to include the back-and-forth that naturally results from that to do it responsibly. [J3] |
| **Intuition (Verifying preprints)** | Journalists relied on trust and intuition as a substitute for, or an addition to, other preprint verification strategies; this gut instinct was viewed critically by some journalists | I just come back to that idea that it is so much about the individual reporter's gut feeling about something. That is, I think, a little scary. Fortunately we have a lot of good reporters working on these things, but, but yeah I don't think that anybody has like a framework that's agreed upon for how to approach these things. [J4] |

(*Continued*)

**Table 3.** (Continued)

| Theme | Definition | Example quotes |
|---|---|---|
| **State that research is unreviewed (Communicating preprints)** | Journalists felt it was important to disclose the unreviewed nature of the preprints they cited (e.g., by labeling it a preprint, stating it had not yet been peer reviewed) | people can share these articles on social networks and everywhere like they're peer-reviewed–like they're something that's already textbook knowledge, which is far from it. This is something that should also be highlighted in the article, and I try to highlight it: 'It's a preprint. It's not peer reviewed yet' [J12] |
| **Contextualize (Communicating preprints)** | Journalists added context to preprint findings by comparing them with information from other sources | I think contextualized properly, they're a really useful and valid source of reporting. . . where possible, you should try and bolster it with other information [J9] |
| **Highlight limitations (Communicating preprints)** | Journalists emphasized the importance of describing caveats, weaknesses, and limitations associated with the preprints they cited | Something that I find fantastic in a lot of medical articles that are hardly found in any other discipline is discussion on limitations, which we are trying to include more and more in our articles [J1] |
| **RQ3: How has the COVID-19 pandemic affected journalists' use of preprints?** | | |
| **A new normal** | COVID-19 preprint coverage was viewed as a complete paradigm shift in science journalism, one likely to continue post-pandemic. | we didn't consider them as really newsworthy items before Covid. Now, we consider them. . . like–something that should be covered like a normal peer-reviewed article, which is a complete paradigm shift, maybe, in science covering [J12] |
| **A moderate shift** | COVID-19 preprint coverage was viewed as a more temporary change in journalism practice, an exception caused by the pandemic | I think that among myself and sort of my friends/peer group you know I think we're pulling back a little bit and I doubt that arXiv is the place a lot of medical reporters are going to eagerly pull reporting from [J4] |
| **Undecided** | Journalists unsure whether preprint news coverage would persist post-pandemic. | It will be interesting to see like, what the implications of that are going forward. . .are preprints going to be covered more generally even outside such an urgent context? [J3]. |

preprints were described as something that should only be reported on if the potential benefits for readers outweighed possible risks of early coverage. This sentiment is captured by statements such as, "The calculation is: 'Do we think that the audience needs to hear the story now or can they wait six to eight weeks. . . for the story to be peer reviewed?'" [J1].

The *timely nature* of preprint findings, in comparison to peer-reviewed research, was a key benefit journalists mentioned as influencing this calculation. This was particularly true during the pandemic, as many of the journalists felt that COVID-19 research with relevance to public health should be made available as soon as possible. However, some journalists saw timeliness as a benefit that extended beyond the post-normal COVID-19 context, such as J9, who reported that "preprints feel like science in motion and in creation. . . they're a place to find the kind of dynamics and flux of science." The timely nature of preprint research was also seen as valuable because it offered journalists a competitive "edge" over colleagues who relied on only peer-reviewed research. Yet, timeliness also acted as a barrier, with several journalists noting that tight deadlines prevented them from verifying the results or general empirical integrity of preprints (see below), or in some cases, from covering them at all.

*Accessibility* was also described as a key advantage of preprints, which are freely available, while many peer-reviewed papers are not. Journalists described accessibility as a personal benefit, as it allowed them to find and use research more easily, but also as a societal one: "That's knowledge that it's not, I think, ethical to be only available to rich people. Especially if it was produced in part with tax dollars, then it's unconscionable that only somebody with a lot of money can get to it" [J16]. One journalist said they appreciated that authors of preprints were typically more accessible for interviews than authors of peer reviewed papers, presumably because they were more excited to discuss their work while the "study is fresh" [J7]. However, this was not a perspective mentioned by other participants.

Journalists considered these anticipated benefits alongside several potential risks. Chief among these was the *potential for preprints to misinform*, a risk many journalists noted had

become particularly relevant during the pandemic. For example, J4 described how the conversation around COVID-19 and schools had become "extremely muddied by preprints," while J16 recalled the challenge of reporting on a vaccine-related preprint in the context of "anti-vax folks" who might misuse the evidence to promote their own agendas.

This potential to misinform was closely linked to the *challenge of verification*, which many of the journalists noted was a barrier to their use of preprints. Journalists described that they did not possess the expertise needed to assess the reliability or accuracy of the research—a challenge shared by both journalists with an advanced degree in a Science, Technology, Engineering, or Math (STEM) field and those without. Journalists noted that preprints could change considerably between coverage and publication in an academic journal, that some may never be published at all, and that it was difficult to tell the difference between which would pass the scrutiny of peer review. Although journalists attempted to mitigate this risk with a number of verification strategies (outlined in the next section), the challenge of verification remained front of mind: "we all want to believe that we can tell what a good preprint is, from what a bad preprint is, and I don't always think that that is true" [J4].

Perhaps as a result of their different risk-benefit calculations, journalists varied widely in the degree to which they supported using preprint research. Some were generally apprehensive, reporting that they felt it was "best to not use them" [J15]. Others were positive, noting that "preprints are now a reality. Everybody can access it. Everybody shares them" [J12]. Still others landed somewhere in the middle, stating that coverage of preprints was acceptable but only "when handled with caution" [J19]. We discuss some of these "cautious" reporting practices in the following section.

## RQ2. What practices do journalists use to find, verify, and communicate the preprints they cover?

Equally varied as the risks and benefits influencing the decision to cover preprints were the strategies journalists used to *find*, *verify*, and *communicate* the research in their stories.

Some journalists relied on active strategies to find preprints, such as J3, who reported "us [ing] preprint repositories like arXiv and medRxiv" during the pandemic. In some cases, these active strategies were contrasted with journalists' "normal" sourcing practices, which were often more passive (e.g. receiving press releases). Still, some journalists discovered preprints in similarly *passive ways*, such as through press releases, authors who mentioned them during interviews, or through services such as the UK's Science Media Centre [54], which releases round ups of expert commentary on new research studies, including preprints. Although journalists sometimes welcomed these passive strategies for finding preprints, they were more often treated with skepticism, "because they can get a little bit promotional" [J7].

When it came to verification, almost all of the journalists said that an additional level of scrutiny was required to vet preprints than to vet peer-reviewed journal publications. This belief was closely tied to the perceived value of the peer-review process, which journalists viewed as a "safety net, basically, that the research has gone through" [J8]. Interestingly, this extra level of skepticism seldom required the use of new, post-normal verification practices, but was instead described as adherence to standard science journalism best practices. As J16 summarized:

> We're using the same toolkit, our toolkit for looking at a paper and evaluating its newsworthiness. We're like, 'Okay. Well, is this a good paper? Is the science good? Do the statistics make sense to us? Do the results actually answer the question that it says it answers, and what's left out?" We ask those things of formally-published—you know, if it comes out in

*Science.* We ask those questions, too, because sometimes the answer is 'No' and sometimes the answer is like, 'Actually, this does seem dicey.'

However, some journalists suggested that they took shortcuts when using non-preprints and deadlines loomed, essentially allowing the peer-review process to replace some of the best practices used to verify information. In addition, a minority of journalists did not explicitly mention treating preprints with extra caution. One such journalist ended up questioning their lack of skepticism during the interview process, reflecting:

I've never really thought of [preprints] as a bad thing. And when you're investigating it in a way that you are, it makes me wonder whether there is a lot of manipulation of the facts going on. So hopefully your study when you report it will give me a few extra clues of what I should be watching for. [J7]

The practices journalists described using to verify preprints included *critically reading* the methods and results, *triangulating* findings with those from other, ideally peer-reviewed stud- ies, and *relying on outside expertise* in a process that resembled scholarly peer review:

Sometimes you need help from other people, say, and you gotta take the study, email it to some experts, and say, 'Okay, I'm gonna do my own peer review with some peers, and we're gonna review it.' And it may be faster, it may not take six months, but we're gonna take a day or two and point out some good and bad things on this study [J5].

Still, despite these multiple and varied formal strategies, intuition also played a role in jour- nalists' verification practices. As J4 explained, "I think that, sort of, it does come down to gut feeling—how much you trust the gut feeling of the people you're reading." This reliance on intuition may be linked to the time-sensitive nature of journalistic work, which, as discussed above, sometimes prevented journalists from applying best practices when verifying preprints. While some journalists were critical of colleagues and peers who relied on gut feeling, others saw intuition as an important journalistic tool that could be used to identify studies warranting additional verification. As J8 explained, if a finding "sounds a little too good to be true. . .it might be."

Finally, journalists applied a range of communication practices when covering preprints. Being *transparent about the unreviewed nature of the research* was chief among these practices, either because of journalists' own beliefs or because doing so was mandated by their organiza- tion. For example, journalists made comments such as, "This is something that should also be highlighted in the article, and I try to highlight it: 'It's a preprint. It's not peer reviewed yet,'" (J12). Other journalists went further, adding that any disclosure of preprint status should be accompanied by an explanation of what the term meant: "if I was going to write, 'This is not peer reviewed,' I'd then have to—would spend at least a sentence—explaining why" [J13]. This sentiment was often closely tied to beliefs about the audiences' level of scientific literacy:

saying whether something is or isn't peer-reviewed—specifically for the kind of outlets that I write for which is, like national media, women's magazines, that kind of thing—is that people probably don't understand, like, what the significance of that is. So, if you want to be kind of legit you just have to kind of really spell out what actually happened. [J13]

In addition to this novel journalistic practice, many journalists also emphasized the impor- tance of applying more standard best practices for science reporting, such as *providing context*

and *highlighting study limitations*. Some journalists went so far as to say that they covered preprints "in the exact same way as published papers" [J12], although this perspective was uncommon.

### RQ3. How has the COVID-19 pandemic affected journalists' use of preprints?

Most journalists reported that COVID-19 had changed their use of preprints, although there was variability in the extent of these perceived changes. Some felt that the pandemic had created "a complete paradigm shift" [J12], both in their own work and in that of their peers. These journalists reported that they were not using preprints before COVID-19, but that the pandemic context "made us all just feel like it was normal and okay to be, you know, skeptically reporting on them and, and paying them a lot of attention" [J4]. Others were more moderate in their views, reflecting that they used preprints occasionally but still "don't report on them much" [J18]. A small number of journalists said that their use of preprints had been unchanged by the pandemic, but this was not a typical perspective.

At the heart of this perceived shift were considerations of the audience's needs and of the urgency of the crisis, with journalists offering explanations such as ". . .there was so little information on COVID. . . we need[ed] to stay really on top of this and cover things, just give the people the information that they need right now" [J3]. Yet, the shift was also described as being tied to a parallel shift within science itself. Specifically, journalists felt that "COVID revealed flaws in the publishing system" [J8] and that the pandemic had normalized preprint use among scholars as well. Other journalists believed that the pandemic had changed the quality of preprints themselves:

> I would say prior to COVID. . .when I would come across preprints or a writer would pitch me a preprint, it was a kind of, "We've got this thing early, but it's almost definitely going to be peer reviewed, it's going to be released in 'X' journal three months later". . . they were journal articles in waiting, really. . . I don't think we were saying, 'and this might get thrown out entirely, who knows?' [J9]

Across these varied perspectives, journalists seemed to agree that preprint coverage had been a "net positive" [J8] in the context of the pandemic. However, they differed in the degree to which they believed, or hoped, this change would persist in the long term. Some had started covering preprints on topics other than COVID-19, or noted that even major legacy publications had started using them. Others reported that they were moving away from preprints, and that in "normal times, I probably wouldn't go to preprints" [J19]. Still others were unsure whether preprints would remain important within journalism outside of the urgent pandemic context.

## Discussion

During COVID-19, journalists shifted their professional norms and practices to more readily include preprints. Our findings suggest this departure is likely to continue post-pandemic and expand beyond the use of preprints related to health and biomedical sciences. As part of their reporting, journalists spoke of regularly seeking scientific papers that preceded formal peer review, often using preprint servers like arXiv and medRxiv. Such scholarship appealed to them, as it was timely, cutting edge, and freely available (as opposed to behind a paywall). This shift marked a departure from "normal" science journalism, with journalists and their media organizations becoming increasingly open to citing unreviewed papers, though typically with

definitions of preprints and caveats (e.g., findings needed to be replicated or part of an evolving story).

Within this post-normal context, journalists worked to verify preprints through a process not entirely dissimilar to peer review, though in greatly compressed timeframes. While the typical formal peer-review process might take weeks or months, journalists truncated their verification process to meet publishing deadlines (e.g., hours or days). As part of this ad hoc process, journalists contacted scientists unaffiliated with the research in question and asked for a critique of the work. Journalists also spoke of triangulating findings as a form of verification, and in some respects, their efforts mirrored the best practices called for in standard, or "normal," science journalism. Yet, most journalists still expressed concerns about the use of preprints. At best, they said, preprints offered potentially life-saving information at a time of great need; at worst, they contributed to misinformation among the public. These findings mirror the results of a survey of 633 science journalists from six world regions about their work during COVID-19, which found that 67% of US and Canadian journalists and 69% of European and Russian journalists had adopted different procedures to cover preprints during the pandemic [19].

Journalists also reported concern that their audiences might misunderstand what preprints are, a concern they attempted to address by offering definitions and context in their articles. Again, this aligns with findings from surveys by Massarani and colleagues [19, 55]. However, this effort stands in contrast to those of recent content analyses, which found that journalists inconsistently identify scholarship mentioned in news articles as preprints, often describing it as "research" or simply hyperlinking to it [13, 41], and only sometimes explain what the term "preprint" means [14]. Further research into a potential disconnect between what professional practices journalists believe they do and what practices actually appear in their published work is needed.

Our findings suggest that there can be value in sharing a preprint version of one's work with journalists, particularly if it regards urgent matters of public health. Preprints offer scientists a more timely and direct way to share information with the public, and journalists can augment the process by offering context and clarification to the research, as well as reach and distribution. Yet, journalists also make errors and frame research in ways at odds with scientists' goals and views [56, 57]. As such, scientists need to weigh the risks and benefits of sharing research through journalists, who act as mediators between them and the public [58].

Both scientists and journalists share responsibility in accurately communicating research that has yet to complete the peer-review process, especially if methodological errors or misinterpretation could have grave public consequences [59, 60]. Most journalists reported not having the expertise to verify the quality of unreviewed research and spoke of the need for scientists to help them vet each study and consider its place in a larger scientific context. Scientists could help address such issues and support journalists in finding, verifying, and accurately communicating their work by understanding the deadline constraints journalists face and avoiding jargon or hype, particularly within the methodology, findings, and limitations sections. Providing lay summaries of preprint findings could also be helpful, particularly for topics that have important public implications [60]. If called on by a journalist to discuss a preprint, researchers may also recommend peers who have the appropriate expertise to vet the findings. Interprofessional training programs in which journalists and scientists learn together to communicate research in a post-normal context could also help maximize benefits and mitigate risks associated with preprint-based media coverage, both during the COVID-19 pandemic and beyond.

Based on our findings, journalists would likely appreciate scientists taking on the role of educator or explainer [61] and allowing time for both interviews and potential follow-up

questions. As verifying preprints is a challenge for journalists, scientists should understand that being asked to comment on a peer's work is, to a large extent, joining the journalist's efforts in orchestrating a truncated peer review [62]. These scientists are being asked to play a key role in shaping whether and how that preprint is covered; their commentary does more than simply add context or legitimize research, as it would in a "normal" science journalism context [32]. As such, scientists asked to comment on preprint findings for a news story should consider the significance and implications of the research with the needs of the public in mind, noting, in particular, any risks that could be associated with the findings. Supporting this vetting process is particularly important, as other practices journalists used to cover unreviewed studies, such as describing them as "preprints," appear to have limited effects on audience perceptions [63, 64].

Scholarly publishers and preprint servers can also support this vetting process by standardizing efforts to show markers of credibility that journalists can use to assess new research (e.g., what, if any, review has taken place; who are the authors, institutions, and funders; what are the potential conflicts of interest, etc. [65]). As the use of preprints has become more commonplace even outside a crisis such as COVID-19, publishers, universities, and other groups with marketing and communication efforts may need to rethink their approach to promoting, an increasingly common activity [66, 67]. For instance, PR efforts could include additional context, links to related evidence, and recommendations for unaffiliated researchers with related expertise on a topic. In addition, journals may wish to revisit policies that encourage preprint use but simultaneously prevent researchers from discussing findings until manuscripts have been formally peer reviewed. Such policies—meant to ensure media attention focuses on the peer reviewed version rather than the preprint—may instead lead to media coverage of preprint research that is not well-vetted and is poorly contextualized [68].

Our findings suggest that although many journalists work behind the scenes to verify, clarify, and communicate the research they cite, these practices are not well established and vary greatly across journalists and outlets. Several journalists expressed concern about the heavy reliance on "gut instinct" in how they and their peers covered preprints. Although some mentioned that their organizations had explicit guidelines about how to report preprints, we could not find any of these online. Organizations such as the Associated Press give a nod to handling preprints in their style guides, advising "extreme care" in their use [6]; however, they fall short in spelling out how to practice such care or how to handle fallout when having reported a preprint that is later discredited or largely changed by the peer-review process. While professional journalism resources are now beginning to recommend some of the post-normal practices identified in this study—such as "commissioning" one's own peer review [69]—concrete guidelines for covering preprints are still far and few between. Journalism associations may seek to address this gap by joining recent efforts to further develop resources and style guides for covering preprints [35, 36, 70]. Professional training and development for journalists, either through universities or continuing education, could also provide additional support.

### Limitations and future directions

Our own backgrounds, as with all qualitative description, shaped this analysis—both as a limitation and a strength. A former journalist (KF) led the interviews, which may have influenced participants' responses. We conducted research at a time of relative stability during the pandemic—the initial vaccine rollout had been completed and boosters were being administered in the US, Canada, and the UK, where most of the journalists were based. It is likely that the views in this paper would differ from those of journalists interviewed at the very beginning of the pandemic. However, the timing allows for us to link the changing practices and norms of

journalists to the changing (i.e., post-normal) communication context. Still, the pandemic remained very much a concern during the time of data collection and publication, with variants of COVID-19 spreading (i.e., Delta, Omicron) and creating uncertainty.

In terms of our sample, all publications in the data set were text-based (not multimedia), English only, science and health news-focused, and based in the Global North. Future research could expand outside these categories, especially given that journalistic preprint use appears to differ across geographic regions [19]. Additionally, although we included niche publishing models (i.e., HealthDay and News Medical), these models remained underrepresented in our sample. HealthDay, for instance, specialized in producing what it called "evidence-based health content" [71] to license to media companies (e.g., CNBC, U.S. News & World Reports, WebMD), hospitals, managed care organizations, publishers, nonprofits, and government agencies. However, email requests for interviews to 8 journalists with the organization went unanswered. While we cannot know why our requests were ignored, HealthDay editors and reporters may have their own norms and practices that are different from those of the journalists we interviewed. With the changing media landscape and broadening definition of "journalist" [44], more research is needed to understand differences and shared norms and practices of journalists at diverse types of outlets, including those at the "margins" of more traditional, legacy journalism [72, 73].

## Conclusions

Collectively, these findings contribute to a still emerging post-normal science communication context that will require new norms and practices for journalists, and perhaps, for the scientists whose work they cite. Our research provides insight into some of these novel journalism practices and the extent to which more established norms for how research is covered have shifted due to the COVID-19 pandemic. The findings fill a gap in our current understanding of how journalists find, vet, and communicate preprints. They build on our previous work [13], which considered journalists' empirical use of research but overlooked other, less visible practices that journalists use to communicate research. The results also act as a reminder that all science is provisional—not just preprints—and that many journalists seem to recognize and communicate this to their audiences.

This research contributes to theory building by using the theoretical framework of post-normal science communication within the emerging COVID-19 context. To a certain extent, our findings align with existing scholarship that has examined post-normal science communication in other contexts, such as climate change. For example, many of the journalists we interviewed strove to put science into context and to communicate the uncertainties associated with preprint research [28–30]. However, we also identified other journalistic practices, such as actively seeking out preprint research and orchestrating one's own peer review, that have not been described in previous studies. Similarly, we found no evidence of journalists advocating for social transformation or engaging audiences in dialogue, both of which are practices that have been associated with post-normal science communication in previous scholarship [20, 28, 30, 31]. More research is needed to better understand why journalists adopt certain communication practices in some post-normal contexts and not others, and what issues these novel practices might raise for their audiences. This is particularly important given that, as Funtowicz and Ravetz [74] state, "we are now truly in a Post-Normal age. Science (and Society) will never be the same again" (p. 3).

## Acknowledgments

We would like to acknowledge Mr. Asura Enkhbayar for his support in collecting the news stories which were used to identify potential participants for the study.

**Disclaimer:** The views expressed in this article are those of the authors and do not necessarily reflect the official policy or position of the Uniformed Services University of the Health Sciences, the Department of Defense, or the U.S. Government.

## Author Contributions

**Conceptualization:** Alice Fleerackers, Laura L. Moorhead, Lauren A. Maggio.

**Data curation:** Alice Fleerackers, Laura L. Moorhead, Lauren A. Maggio, Kaylee Fagan, Juan Pablo Alperin.

**Formal analysis:** Alice Fleerackers, Laura L. Moorhead, Lauren A. Maggio, Kaylee Fagan, Juan Pablo Alperin.

**Funding acquisition:** Alice Fleerackers, Laura L. Moorhead, Lauren A. Maggio, Juan Pablo Alperin.

**Investigation:** Alice Fleerackers, Laura L. Moorhead, Lauren A. Maggio, Juan Pablo Alperin.

**Methodology:** Alice Fleerackers, Laura L. Moorhead, Lauren A. Maggio.

**Project administration:** Alice Fleerackers, Laura L. Moorhead, Juan Pablo Alperin.

**Resources:** Laura L. Moorhead, Juan Pablo Alperin.

**Supervision:** Laura L. Moorhead.

**Writing – original draft:** Alice Fleerackers, Laura L. Moorhead, Lauren A. Maggio, Kaylee Fagan, Juan Pablo Alperin.

**Writing – review & editing:** Alice Fleerackers, Laura L. Moorhead, Lauren A. Maggio, Kaylee Fagan, Juan Pablo Alperin.

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
