## [Decision Letter · Decision Letter 0]

7 Oct 2022

PONE-D-22-04088Science in motion: A qualitative analysis of journalists' use and perception of preprintsPLOS ONE

Dear Dr. Fleerackers,

Thank you for submitting your manuscript to PLOS ONE. After careful consideration, we feel that it has merit but does not fully meet PLOS ONE’s publication criteria as it currently stands. Therefore, we invite you to submit a revised version of the manuscript that addresses the points raised during the review process. Please submit your revised manuscript by Nov 11 2022 11:59PM. If you will need more time than this to complete your revisions, please reply to this message or contact the journal office at plosone@plos.org. Please include the following items when submitting your revised manuscript:

We look forward to receiving your revised manuscript.

Kind regards,

Claudia Noemi González Brambila, Ph.D.

Academic Editor

PLOS ONE

Journal Requirements:

“This research is supported by a Social Sciences and Humanities Research Council of Canada (SSHRC) Insight Grant (#453-2020-0401; JPA) and a SSHRC Joseph Bombardier Doctoral Fellowship (#767-2019-0369; AF). The funders had no role in study design, data collection and analysis, decision to publish, or preparation of the manuscript.”

[“This research is supported by the Social Sciences and Humanities Research Council of Canada (SSHRC), grant number 453-2020-0401. AF is supported by a SSHRC Joseph Bombardier Doctoral Fellowship, grant number 767-2019-0369.”

“This research is supported by a Social Sciences and Humanities Research Council of Canada (SSHRC) Insight Grant (#453-2020-0401; JPA) and a SSHRC Joseph Bombardier Doctoral Fellowship (#767-2019-0369; AF). The funders had no role in study design, data collection and analysis, decision to publish, or preparation of the manuscript.”

5.Please review your reference list to ensure that it is complete and correct. If you have cited papers that have been retracted, please include the rationale for doing so in the manuscript text, or remove these references and replace them with relevant current references. Any changes to the reference list should be mentioned in the rebuttal letter that accompanies your revised manuscript. If you need to cite a retracted article, indicate the article’s retracted status in the References list and also include a citation and full reference for the retraction notice.

Reviewers' comments:

Reviewer's Responses to Questions

**Comments to the Author**

1. Is the manuscript technically sound, and do the data support the conclusions?

Reviewer #1: Yes

Reviewer #2: Partly

2. Has the statistical analysis been performed appropriately and rigorously? 

Reviewer #1: Yes

Reviewer #2: I Don't Know

3. Have the authors made all data underlying the findings in their manuscript fully available?

Reviewer #1: Yes

Reviewer #2: No

4. Is the manuscript presented in an intelligible fashion and written in standard English?

Reviewer #1: Yes

Reviewer #2: Yes

5. Review Comments to the Author

Reviewer #1: The paper addresses a new topic that needs to be addressed.

The paper offers an intensive study that does add a contribution to the knowledge of preprint research as a form of post-normal science. Therefore, the results are new.

Results are presented clearly and analysed appropriately

The work has a novelty and originality

The significance, methodology and objectives of the article are explained clearly

The references are recent and convincing

More recommendations should be suggested

Reviewer #2: I have reviewed the paper “Science in motion: A qualitative analysis of journalists' use and perception of preprints”. I found the paper interesting, however, it is my impression that in its current form, it has some limitations that need to be addressed before it is ready to be published.

Authors need to include in their analysis not only the agreement between the opinions of the 19 journalists interviewed but also the discrepancies that exist between them. Reading the document, both in Table 3 and in the description on page 17, it appears that they all agree or have very similar points on the three research questions, which seems unlikely given their different academic and professional backgrounds and the variety of topics covered in the interviews.

My second main concern is about the contribution of the paper. Authors need to be clearer about why is it important that we know what they are proposing, there does not seem to be a clear discussion with other authors about their findings, from the literature review conducted, do their results prove or contradict what has been said by any other author(s)?

Moreover, the conclusions also have an opportunity to be improved. Authors could make suggestions about the key challenges and issues that post-normal science communication would face in the post-COVID-19 pandemic.

Hope the authors find my comments helpful to improve the paper.

6. PLOS authors have the option to publish the peer review history of their article (what does this mean?). If published, this will include your full peer review and any attached files.

Reviewer #1: **Yes: **Dr. Afaf Abu Sirhan

Reviewer #2: No

---

## [Author Response · Author response to Decision Letter 0]

14 Oct 2022

Please see Response to Reviewers letter (attached)

---

## [Decision Letter · Decision Letter 1]

3 Nov 2022

Science in motion: A qualitative analysis of journalists' use and perception of preprints

PONE-D-22-04088R1

Dear Dr. Fleerackers,

We’re pleased to inform you that your manuscript has been judged scientifically suitable for publication and will be formally accepted for publication once it meets all outstanding technical requirements.

Kind regards,

Claudia Noemi González Brambila, Ph.D.

Academic Editor

PLOS ONE

Additional Editor Comments (optional):

Reviewers' comments:

Reviewer's Responses to Questions

**Comments to the Author**

1. If the authors have adequately addressed your comments raised in a previous round of review and you feel that this manuscript is now acceptable for publication, you may indicate that here to bypass the “Comments to the Author” section, enter your conflict of interest statement in the “Confidential to Editor” section, and submit your "Accept" recommendation.

Reviewer #1: All comments have been addressed

Reviewer #2: All comments have been addressed

2. Is the manuscript technically sound, and do the data support the conclusions?

Reviewer #1: Yes

Reviewer #2: Yes

3. Has the statistical analysis been performed appropriately and rigorously? 

Reviewer #1: Yes

Reviewer #2: N/A

4. Have the authors made all data underlying the findings in their manuscript fully available?

Reviewer #1: Yes

Reviewer #2: No

5. Is the manuscript presented in an intelligible fashion and written in standard English?

Reviewer #1: Yes

Reviewer #2: Yes

6. Review Comments to the Author

Reviewer #1: Dear author

thank you for response letter,

be aware that the numbering of lines is not correct in the response table, but it is OK

Reviewer #2: The corrections made to the document make it acceptable for publication without any additional modifications.

7. PLOS authors have the option to publish the peer review history of their article (what does this mean?). If published, this will include your full peer review and any attached files.

Reviewer #1: **Yes: **Afaf A. Abu Sirhan

Reviewer #2: No

---

## [Editor Report · Acceptance letter]

10 Nov 2022

PONE-D-22-04088R1 

Science in motion: A qualitative analysis of journalists' use and perception of preprints 

Dear Dr. Fleerackers:

I'm pleased to inform you that your manuscript has been deemed suitable for publication in PLOS ONE. Congratulations! Your manuscript is now with our production department. 

Kind regards, 

on behalf of

Dr. Claudia Noemi González Brambila 

Academic Editor

PLOS ONE